# Automated Optimization-Based Deep Learning Models for Image Classification Tasks

Daudi Mashauri Migayo [1,2,*], Shubi Kaijage [1], Stephen Swetala [3] and Devotha G. Nyambo [1]

1    School of Computational and Communicational Science and Engineering, The Nelson Mandela African Institution of Science and Technology, Arusha 23311, Tanzania; shubi.kaijage@nm-aist.ac.tz (S.K.); devotha.nyambo@nm-aist.ac.tz (D.G.N.)
2    Department of Business Administration, Tanzania Institute of Accountancy (TIA), Dar es Salaam 15108, Tanzania
3    Department of Orthopedic and Trauma Surgery, Bugando Medical Centre, Mwanza 33102, Tanzania; stephenswetala@gmail.com
*    Correspondence: migayod@nm-aist.ac.tz; Tel.: +255-786-073-629

**Abstract:** Applying deep learning models requires design and optimization when solving multi-faceted artificial intelligence tasks. Optimization relies on human expertise and is achieved only with great exertion. The current literature concentrates on automating design; optimization needs more attention. Similarly, most existing optimization libraries focus on other machine learning tasks rather than image classification. For this reason, an automated optimization scheme of deep learning models for image classification tasks is proposed in this paper. A sequential-model-based optimization algorithm was used to implement the proposed method. Four deep learning models, a transformer-based model, and standard datasets for image classification challenges were employed in the experiments. Through empirical evaluations, this paper demonstrates that the proposed scheme improves the performance of deep learning models. Specifically, for a Virtual Geometry Group (VGG-16), accuracy was heightened from 0.937 to 0.983, signifying a 73% relative error rate drop within an hour of automated optimization. Similarly, training-related parameter values are proposed to improve the performance of deep learning models. The scheme can be extended to automate the optimization of transformer-based models. The insights from this study may assist efforts to provide full access to the building and optimization of DL models, even for amateurs.

**Keywords:** deep learning; automated optimization; automated machine learning; parameter optimization; sequential-model-based optimization; image classification





## 1. Introduction

Artificial intelligence (AI) generally involves modeling intelligent behavior using a computer program with limited human interference [1]. AI manifests through numerous applications, such as autonomous driving, natural language processing, intelligent information retrieval, expert consulting systems, theorem proving systems, robotics, automatic programming, combinatorial and scheduling systems, and perception problems [2,3].

Machine learning (ML) is considered a subfield of AI [4]. However, some believe that only the intelligent part of ML should be regarded as a subset of AI [5]. Either way, both kinds of literature converge on the notion that in ML a computational algorithm is built that can be applied to make decisions or estimates based on training data, without being explicitly programmed to perform the task [4,6].

An ML model learns from experience for a given task if its performance towards the task improves with experience [7]. Machine learning models are primarily classified into four groups: supervised learning, unsupervised learning, semi-supervised learning, and reinforcement learning [8]. Deep learning leads to other machine learning tools in general imaging and computer vision [9].

Deep learning (DL) refers to techniques that build on artificial neural networks in which multiple network layers are added to increase the levels of abstraction and performance [10]. For better generalizability, a good DL model is obtained through several processes: pre-processing of data, feature engineering, model generation, and model evaluation [11]. Model generation includes the design of DL models and optimization during training. Model generation is turned into an optimization process for state-of-the-art (SOTA) DL models with transfer learning.

The optimization process can either involve hyperparameter optimization (HPO) or architecture optimization (AO) [11]. The former focuses on tuning training-related parameters, for example, the batch size, learning rate, and the number of training iterations. The latter deals with model-related parameters, for example, the number of hidden layers, filter size, and the number of neurons per layer. Popular optimization methods include grid and random search [12,13], Bayesian optimization [14,15], and gradient-based optimization [16–19].

Standard DL models require determining both training- and model-related parameters before training. The model's performance can be affected considerably by the choice of these parameters, yet finding good values is notably tough [20]. The problem is exacerbated by the fact that some parameter values are integers (e.g., batch size), others are floating-point (e.g., learning rate), and others are categorical (e.g., optimizer). During training, landing on the veracious set of parameter values for better generalization remains an indistinct procedure, hampering the replication of ML experiments.

Luckily, there has been growing interest in automating the ML pipeline, i.e., automated machine learning (AutoML), to free data scientists from burdensome tasks. Google introduced publicly shared Cloud AutoML (https://cloud.google.com/automl/, accessed on 4 November 2022) systems, and others have submitted open-source optimization libraries, i.e., Hyperopt [21], Skopt [22], SMAC [23], and KerasTuner (https://keras.io/keras_tuner/, accessed on 29 June 2023). Similarly, the majority of recent literature on AutoML for DL models concentrates on automating the design of DL models through the neural architecture search (NAS) process [24–28]. However, works on automating the optimization process after obtaining the candidate architecture through NAS is still insufficient.

To date, an insufficient number of works of literature on the automated optimization of DL models have been identified. Similarly, building high-quality DL models through the optimization process can take time for non-experts. Therefore, existing difficulties motivated a desire to seek improved approaches for model optimization. Deliverables from such systems may be applied to join efforts to free data scientists from the burdensome optimization task.

For that reason, this empirical work proposes to advance the understanding of this growing area of research by introducing automated optimization of DL models for computer vision, specifically image classification tasks. Four DL models, each presenting a different architecture and prominent datasets, were employed during the implementation of the proposed solution. The proposed solution experimented with a transformer-based model to test its applicability to vision transformer models.

Data scientists, DL practitioners, experts, and non-experts working on image classification tasks can use the proposed solution without geographical limitations. A data scientist has control over the optimization attempts in the proposed solution, as opposed to using brute force and exhaustive searches. The main contributions of this paper can be summarized as follows:

- Through a series of empirical tests, the association and impact on the performance of two main training-related parameters, batch size and learning rate, is explored;
- A framework for automated optimization of the DL model for image classification tasks is presented;
- An empirical demonstration showing that the proposed framework improves the performance of DL models is presented;

- A set of training-related parameter values for better performance of DL models for further extensive empirical evaluations is recommended.

The remainder of this paper is structured as follows: Section 2 concerns materials and methods. Section 3 is dedicated to presenting results; subsequently, Section 4 discusses results and findings. Section 5 outlines a conclusion and some highlights of what to expect in future works.

## 2. Materials and Methods

This section summarizes the experimental setup, presents an overview of selected deep learning models, parameters to be used during automated optimization, datasets, and performance measures for empirical evaluation; an overview of the problem at hand is provided, proceeded by an optimization algorithm and the proposed framework.

### 2.1. Experimental Setup

Three sets of experiments were conducted in this study: the first to assess the association of learning rate and batch size; the second to assess the impact of optimizers; and the third to demonstrate the proposed solution. The first set involved twenty tests, the second set involved three, and the third involved three tests for each of the selected DL models. A total of 104 experimental runs were conducted.

All experiments were coded using Python and run using an online Jupyter Notebook executed on the Google Colab cloud computing on a TensorFlow framework. A graphics processing unit (GPU) offered on the Google platform was utilized to expedite the training process. At the beginning of each training session, Google offers up to 12.7 GB of RAM, 15 GB of GPU, and 78.2 GB of disk space that are dynamically assigned and utilized.

### 2.2. Deep Learning Models

Four of the top state-of-the-art deep convolution neural network (D-CNN) models, standard for image classification tasks, were selected in this work. A very deep convolutional network for large-scale image recognition (VGG-16) is beneficial as it generalizes well on other databases, apart from the ImageNet database, while achieving SOTA results [29]. GoogleLeNet or Inception uses fewer computational resources and few parameters while achieving high SOTA results [30]. In this paper, we considered Inception version V3.

Likewise, from the clan of Residual Network (ResNet), ResNet50 was selected as it presents a novel architecture, skipping connections that go deeper by up to 152 layers (8 times deeper than VGG-19) but still maintaining low complexity [31]. The final nomination was the EfficientNet, which introduced a novel model scaling method called compound scaling [32]. The proposed approach broke the mold of its predecessors, which previously used the conventional system of stacking many layers to scale up the model. EfficientNetB0 was considered in these experiments.

The selected four DL models are winners of the ImageNet Large Scale Visual Recognition Challenge (ILSVRC) at different timeframes. In the ILSVRC 2014, VGG-16 was ranked second, and Inception was first. Likewise, ResNet50 was ranked first in the ILSVCR 2015. Lastly, EfficientNet was a winner in the ILSVRC 2019.

Each of the selected models presents a unique architecture that generalizes well on a variety of datasets (VGG-16), minimizes computational resources (Inception), gets deeper while reducing complexity (ResNet50), and uniformly scales up (EfficientNet). Diverse architectures in the selected models lays a foundation for most variants of D-CNN.

A transformer-based model by Dosovitskiy et al. [33] was used to test the applicability of the proposed approach to vision transformers (ViT). The model contains multiple transformer blocks with a multi-head attention layer applied to the sequence of image patches. The final output of each transformer block is flattened to be used as the representation of image input to the classifier head. The classifier head uses a softmax activation to produce the output probability of each class.

In this paper, deep learning models based on ConvNets were mainly considered in the experiments. However, in recent years, the popularity of transformer-based models for computer vision applications has been growing. Yet recent ConvNets can be as robust and reliable or, in some cases, even more so than transformers [34]. Therefore, the experiments and results presented in this paper primarily concentrate on ConvNet-based deep learning models; however, at the same time, the proposed approach was evaluated with a transformer-based model.

### 2.3. Parameters

Four parameters were used in performing automated optimization during the empirical work: the number of trainable hidden layers, learning rate, batch size, and optimizer. Nevertheless, the number of training iterations (epoch) remained the same because early stopping was used in the selected DL models. Early stopping monitored improvements in the validation loss with patience set to 10 and mode set to min, as we seek to minimize loss when validation accuracy improves, restoring best model weights.

### 2.4. Datasets and Performance Measures

Standard datasets for image classification tasks used across research and hackathons were selected [35,36]. The ImageNet (https://image-net.org/download, accessed on 4 November 2022) is the largest dataset for image classification and localization. In addition, we selected Fashion-MNIST [37], Stanford Cars [38], and the Google Cats and Dogs dataset (https://storage.googleapis.com/mledu-datasets/cats_and_dogs_filtered.zip, accessed on 4 November 2022). All datasets are publicly available and can be freely accessed for various image classification challenges.

With a balanced dataset used to evaluate the proposed approach, the performance metric used in this work is accuracy. Accuracy measures the probability of an image being classified as positive or negative. With initial accuracy $Acc_i$ and final accuracy $Acc_f$, the relative error $\varepsilon_r$ drop can be calculated as:

$$\varepsilon_r = \frac{Acc_f - Acc_i}{100 - Acc_i} \times 100\% \tag{1}$$

Evaluation metrics like precision, recall, and confusion matrix may be employed when the proposed approach uses imbalanced datasets. However, a data scientist must explicitly define the evaluation metric before optimizing.

### 2.5. Problem Definition

This work addresses the following problem. Given:

- A set of deep learning models $M^{(1)},\ldots, M^{(t)}$, with t being the number of deep learning candidate models for a CV task T;
- A number m of datasets as $\mathcal{D}^{(1)},\ldots, \mathcal{D}^{(m)}$;
- A set of n parameters with domains $\Theta_1,\ldots, \Theta_n$;
- Model $M$'s configuration space as a product of parameter domains given as $\Theta = \Theta_1 \times \ldots \times \Theta_n$;
- For each of the selected model $M$ on dataset $\mathcal{D}$ during task T, a number of K sets have a pair of empirical performance metrics $y_i$ with parameter settings $\theta_i$ as $\langle y_i, \theta_i \rangle_{i=1}^{K}$.

Our goal is to automate the process of determining, for the selected model on a given dataset for a particular image classification task, which parameter values are to be applied for the expected better performance of a model during the optimization process.

### 2.6. Optimization Algorithm

We employed a sequential-model-based optimization algorithm to implement the proposed scheme. More specifically, we used Bayesian optimization (BO), a popular SMB-

based hyperparameter optimization method [14,15]. With an evaluation function f and acquisition function S, SMBO can be articulated in Algorithm 1 [14].

---

**Algorithm 1**: Sequential-Model-Based Optimization

---

Input: f, $\Theta$, S, M
$\mathcal{D} \leftarrow$ INITSAMPLES (f, $\Theta$)
**for** i in [1, 2, . . . , t] **do**
$p(y \mid \theta, \mathcal{D}) \leftarrow$ FITMODEL (M, $\mathcal{D}$)
$\theta_i \leftarrow \text{argmax}_{\theta \in \Theta} S(\theta, p(y \mid \theta, \mathcal{D}))$
$y_i \leftarrow f(\theta_i)$
$\mathcal{D} \leftarrow \mathcal{D} \cup (y_i, \theta_i)$
**end for**

---

After tuning the probabilistic model M to fit on $\mathcal{D}$, function S selects subsequent promising neural architecture from M. A balance between exploring new architectures from M and exploiting architectures already identified to have auspicious values is defined by S. Function f evaluates the selected neural architecture after training and validation. The new pair of evaluation results $(y_i, \theta_i)$ is appended on $\mathcal{D}$.

*2.7. Automated Optimization Framework*

The proposed framework of the automated optimization scheme can be summarized in Figure 1. The optimization scheme accepts a DL model, classification task, and control variable to exploit the configuration space. Finally, it yields the best-performing DL model, associate parameters, and evaluation metrics.

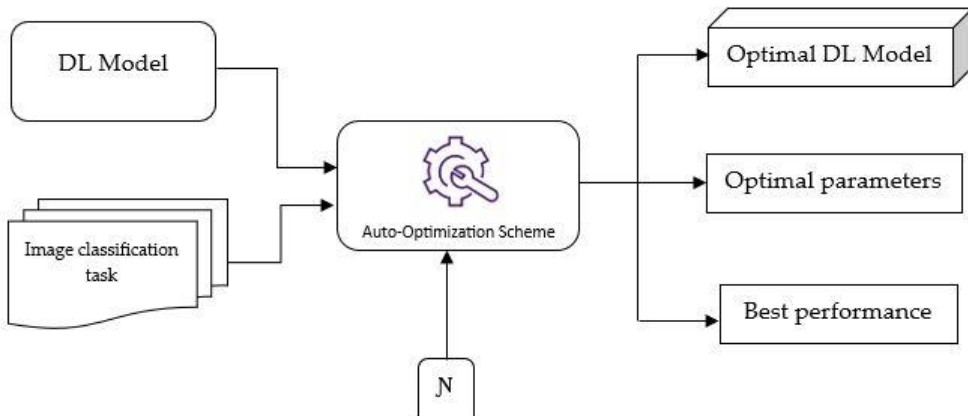

**Figure 1.** Automated optimization framework.

At first, a data scientist specifies a DL model M, dataset $\mathcal{D}$, and a maximum number of optimizations attempt $\mathcal{N}$ for a given image classification task. The algorithm compiles a model after preprocessing the data on $\mathcal{D}$ using default parameters $\theta_1$. Afterwards, the algorithm fits M on $\mathcal{D}$ and records the initial performance measure $y_1$ using $\theta_1$. A comparison between $y_1$ and new performance values will be made during the optimization process.

The algorithm then initializes automated optimization of M on $\mathcal{D}$ using configuration space $\Theta$. Parameters $\theta_i$ are selected iteratively to compile and fit M on $\mathcal{D}$ while recording new performance $y_i$. If M improves as $y_i$ is compared with $y_1$, the algorithm updates the performance measure and corresponding parameter value $(y_i, \theta_i)$ on saving M. Otherwise, the algorithm retrains after unfreezing the top layers of M and reducing the learning rate. In this paper, unfreezing happened layer by layer, since the training process of the internal layers of DL differs significantly [39].

In both cases, if there are no improvements, parameters $\theta_i$ and compiled M are discarded upon selecting subsequent parameter values for the next optimization attempt. The counter for optimization attempts, t, will be incremented. The process repeats with

$\mathcal{N}$ controlling the exploration of $\theta_i$ from $\Theta$ when exploiting selected parameters. The automated optimization can be summarized in Algorithm 2.

---

**Algorithm 2**: Automated Optimization of DL Models

---

**Input**: M, $\mathcal{D}$, $\mathcal{N}$
**Output**: best M, final $y_i$, and values $\theta_i$
**Method**: the algorithm works as follows.
1: Initialize M on $\mathcal{D}$ with $\theta_1$
2 : $y_1 \leftarrow f(\theta_1)$
3 : record $(y_i, \theta_i)$
4 : **for** $\theta_i$ in $\Theta$ **do**
5 :     $y_i \leftarrow f(\theta_i)$
6 :     **if** $y_i$ improves, update $(y_i, \theta_i)$
7:     **else** unfreeze M
8 :             $y_i \leftarrow f(\theta_i)$
9 :             **if** $y_i$ improves, update $(y_i, \theta_i)$
10 :            **else** discard $\theta_i$, increment t
11:             **end if**
12:         **end if**
13 : **while** t < $\mathcal{N}$, reiterate

---

Most of the existing approaches optimize models for ML tasks other than computer vision. Optimizing a model for a computer vision task is often a resource-intensive task. The proposed method stands out from the existing literature for three reasons. First, it automates the optimization process of models for computer vision, a solution currently insufficiently addressed in the current literature. Second, it can be configured and supports state-of-the-art deep learning models with transfer learning for any image classification task. Third, a user controls the optimization process and may limit the utilization of resources, as opposed to using brute force and exhaustive searches.

### 2.8. Implementation

Transfer learning was employed on the selected models with pre-computed weights on ImageNet. The Fashion-MNIST and Stanford Cars datasets were used to initially examine the association between batch sizes and learning rates during training with a generic model. The Stanford Cars dataset encompasses 16,185 images and 196 car classes. Fashion-MNIST holds 70,000 28 × 28 grayscale fashion images from 10 categories, divided into 60,000 training and 10,000 test examples. The proposed automated optimization approach was implemented and evaluated on the Stanford Cars and the Cats and Dogs datasets with 1500 color images for each class, split into 2000 images for training and 1000 for validation.

Three optimizers were used: RMSProp, SGD with momentum, and Adam. Batch size increases by 2, starting with 8 through to 128. The number of epochs was fixed to 100 with the early stopping monitoring validation loss, with patience set to 10, and a 0.5 dropout rate to prevent overfitting, and saving only the best model.

## 3. Results

This section presents the result of our empirical work before, during, and after building an automated optimization-based deep learning model for computer vision.

### 3.1. Hyperparameter Association

At first, a series of empirical tests were conducted to explore the relationship between batch size and learning rate and their impact on the model's performance. For each of the selected DL models, a pair of batch size and learning rate was iteratively determined while observing accuracy and loss during training and validation. Table 1 presents the experimental data for ResNet50. Epoch represents the number of iterations the activity training terminated using the early stopping criteria. Likewise, the time in minutes shows

the corresponding duration of training. Appendix A provides supplementary experimental data. Table A1 gives experimental data for a VGG-16; Table A2 gives experimental data for Inception; Table A3 gives experimental data for EfficientNet.

**Table 1.** Training data for ResNet50.

| Batch Size | Learning Rate | Loss | Accuracy | Validation Loss | Validation Accuracy | Epoch | Time (min) |
|---|---|---|---|---|---|---|---|
| 8 | 0.1 | 0.8353 | 0.4827 | 0.8761 | 0.4929 | 11 | 7 |
| 16 | 0.1 | 0.7928 | 0.5555 | 0.787 | 0.5167 | 49 | 30 |
| 32 | 0.1 | 0.8922 | 0.4937 | 0.9006 | 0.5151 | 18 | 11 |
| 64 | 0.1 | 0.9066 | 0.4639 | 1.0116 | 0.4712 | 29 | 20 |
| 128 | 0.1 | 1.0363 | 0.4645 | 0.7907 | 0.4675 | 28 | 18 |
| 8 | 0.01 | 0.2277 | 0.8506 | 0.3207 | 0.8331 | 64 | 31 |
| 16 | 0.01 | 0.0214 | 0.9198 | 0.114 | 0.9141 | 25 | 15 |
| 32 | 0.01 | 0.2312 | 0.8553 | 0.5977 | 0.7737 | 40 | 22 |
| 64 | 0.01 | 0.0172 | 0.9277 | 0.0727 | 0.926 | 50 | 29 |
| 128 | 0.01 | 0.0022 | 0.9304 | 0.0603 | 0.9316 | 26 | 14 |
| 8 | 0.001 | 0.0239 | 0.9292 | 0.0686 | 0.9371 | 35 | 13 |
| 16 | 0.001 | 0.0584 | 0.9284 | 0.0827 | 0.9376 | 20 | 12 |
| 32 | 0.001 | 0.0079 | 0.9442 | 0.0927 | 0.9403 | 15 | 9 |
| 64 | 0.001 | 0.0008 | 0.9514 | 0.0786 | 0.9404 | 12 | 7 |
| 128 | 0.001 | 0.005 | 0.9504 | 0.0616 | 0.9383 | 16 | 6 |
| 8 | 0.0001 | 0.6079 | 0.9395 | 0.0615 | 0.9424 | 63 | 45 |
| 16 | 0.0001 | 0.027 | 0.9506 | 0.0865 | 0.9369 | 19 | 12 |
| 32 | 0.0001 | 0.0974 | 0.9439 | 0.1006 | 0.9427 | 78 | 46 |
| 64 | 0.0001 | 0.0409 | 0.9469 | 0.1046 | 0.9458 | 11 | 6 |
| 128 | 0.0001 | 0.0569 | 0.9534 | 0.105 | 0.9539 | 11 | 6 |

The data in Table 2 present the results that indicate a positive association between batch size and learning rate, with the best performance obtained when high rates are used with large mini batches and the inverse is true.

**Table 2.** Association of hyperparameters during training and optimization.

| DL Model | Batch Size | Learning Rate | Validation Accuracy | Epoch |
|---|---|---|---|---|
| ResNet50 | 128 | 0.01 | 98.7% | 26 |
|  | 8 | 0.01 | 88.7% | 66 |
| VGG-16 | 128 | 0.01 | 92.6% | 16 |
|  | 8 | 0.01 | 89.5% | 32 |
| Inception | 128 | 0.01 | 95.4% | 11 |
|  | 8 | 0.01 | 83.4% | 18 |
| EfficientNetB0 | 128 | 0.01 | 95.8% | 14 |
|  | 8 | 0.01 | 60.1% | 21 |

The experimental evidence in Figure 2 reveals that a DL model diverges following training with a low rate and a large mini batch (a). Contrastingly, a clear benefit of using low rates with small mini batches can be witnessed as the model converges to a local optimal solution (b). Nevertheless, the latter's training time is higher than the former.

If we turn to the training losses, a combination of low rates and a small mini batch promises convergence in reducing training and validation loss, as shown in Figure 2d. Conversely, low rates and a large mini batch guarantee training and validation loss divergence, as seen in Figure 2c.

The data in Figure 2 shows that model accuracy and loss improve positively when the model is trained using low rates and a small mini batch. Divergence in model accuracy and loss is observed when the model is trained using low rates and a high mini batch, or the inverse. The following subsection concerns performance gains obtained after applying the proposed automated framework during the training and optimization of DL models.

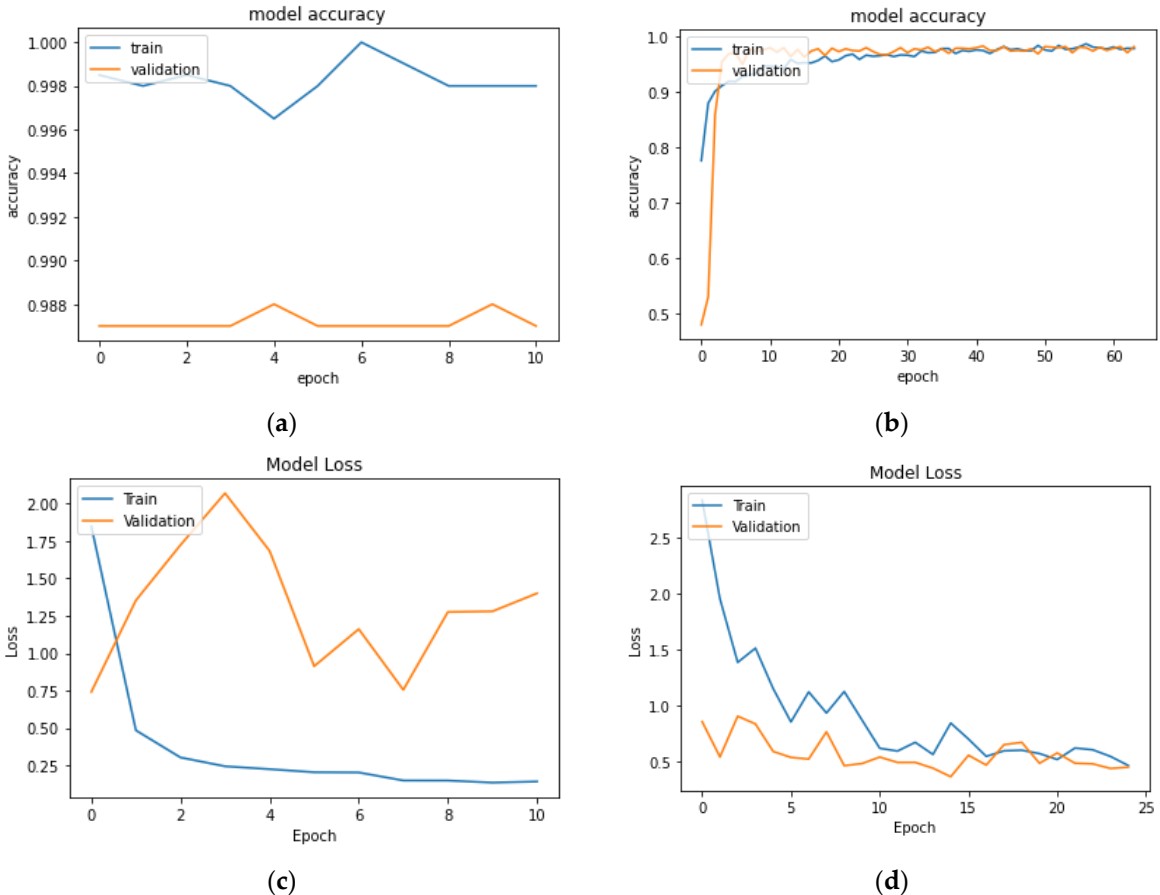

**Figure 2.** Model performance when trained at a rate of 0.0001: (**a**) model diverges with mini batch of 128; (**b**) model converges with a mini batch of 8. Model loss when trained at a rate of 0.001: (**c**) model diverges with a minibatch of 128; (**d**) model converges with minibatch of 8.

### 3.2. Performance Evaluation

Utilizing the proposed framework resulted in performance improvements of selected DL models after training and optimization. As can be seen from the data in Table 3, the error rate was reduced by improving accuracies for the selected DL model. The initial accuracy was obtained through the default training parameters of batch size 32 and a learning rate of 0.001, as suggested by existing literature. The automated optimization process achieved final accuracy after a series of empirical trials.

**Table 3.** Post-automated optimization performance evaluation.

| DL Model | Initial Accuracy | Final Accuracy | Training Time | Batch Size | Learning Rate |
|---|---|---|---|---|---|
| VGG-16 [29] | 0.9370 | 0.9830 | 01:00:58 | 32 | $1 \times 10^{-5}$ |
| Inception [30] | 0.9870 | 0.9920 | 00:41:25 | 16 | $1 \times 10^{-5}$ |
| ResNet50 [31] | 0.9820 | - | 03:17:26 | 32 | $1 \times 10^{-3}$ |
| EfficientNet [32] | 0.9830 | 0.9890 | 02:41:08 | 16 | $1 \times 10^{-5}$ |

Closer inspection of Table 3 reveals an upsurge of accuracy from 0.9370 to 0.9830 for a VGG-16, signifying a 73.02% relative error drop from 0.063 to 0.017 as calculated using Equation (1) in approximately one hour. Similarly, there was a 37.29% relative error reduction for EfficientNet as the accuracy rose from 0.9830 to 0.9890 after utilizing the proposed framework. Contrary to expectations, no performance improvement was observed on ResNet50, as the initial accuracy could not be surpassed. These results indicate

that the proposed framework can automate the optimization process of DL models and, indeed, improve performance by significant proportions.

Table 3 evaluates the optimization process using the proposed scheme regarding accuracy and training time. Accuracy represents model performance after the automated optimization process as an evaluation criterion. Training time means the consumption of resources during the automated optimization process.

### 3.3. Proposed Parameter Values

Several pairs of training-related parameters for better DL model performance were observed through our empirical tests. We recommend these values during the training of DL models for further extensive empirical evaluations. Generally, learning rates of 0.01, 0.001, and 0.0001 work well with mini batches 64, 32, and 16, respectively. However, the learning rate should be reduced if the top layers of a DL model are unfrozen to avoid wrecking precomputed weights. Table 4 summarizes these parameter values.

**Table 4.** A summary of proposed training-related parameters for DL models.

| Batch Size | Learning Rate | Learning Rate (Unfrozen) |
| --- | --- | --- |
| 16 | $1 \times 10^{-4}$ | $1 \times 10^{-5}$ |
| 32 | $1 \times 10^{-3}$ | $1 \times 10^{-4}$ |
| 64 | $1 \times 10^{-2}$ | $1 \times 10^{-3}$ |

These values can be adopted as default parameters while training DL for image classification tasks. Based on our empirical evaluations, these values guarantee the optimal performance of DL models. Data scientists and DL practitioners may select and apply a pair of proposed values when building a baseline model. Afterward, the optimization process to surpass a baseline model may be initiated by utilizing automated tools like the proposed scheme.

### 3.4. Optimizers

Empirical tests were conducted to assess the optimizers' impact on accuracy and loss. A default learning rate of 0.001 and a batch size of 32 were selected during training with varying optimizers. Three optimizers were used: Adam, SGD, and RMSprop. Table 5 provides a summary of the outcomes of these experiments.

**Table 5.** The impact of optimizers on accuracy and loss.

| DL Model | Optimizer | Loss | Accuracy | Validation Loss | Validation Accuracy | Epoch | Time (min) |
| --- | --- | --- | --- | --- | --- | --- | --- |
| | Adam | 0.3317 | 0.8905 | 0.3356 | 0.8720 | 27 | 19 |
| ResNet50 | SGD | 0.0084 | 0.9975 | 0.0641 | 0.9810 | 38 | 22 |
| | RMSprop | 0.8384 | 0.5845 | 0.8203 | 0.4560 | 23 | 15 |
| | Adam | 0.2586 | 0.8855 | 0.1723 | 0.9360 | 36 | 21 |
| VGG-16 | SGD | 0.3987 | 0.8140 | 0.2496 | 0.9020 | 63 | 19 |
| | RMSprop | 0.3057 | 0.8685 | 0.1897 | 0.9290 | 25 | 12 |
| | Adam | 0.1015 | 0.9590 | 0.1101 | 0.9570 | 38 | 11 |
| Inception | SGD | 0.1064 | 0.9540 | 0.0975 | 0.9620 | 21 | 6 |
| | RMSprop | 0.1283 | 0.9555 | 0.0955 | 0.9620 | 38 | 10 |
| | Adam | 0.1217 | 0.9640 | 0.0780 | 0.9810 | 39 | 19 |
| EfficientNet | SGD | 0.0789 | 0.9690 | 0.0458 | 0.9840 | 18 | 9 |
| | RMSprop | 0.2129 | 0.9665 | 0.2235 | 0.9800 | 31 | 15 |

From the data in Table 5, except for VGG-16, the rest of the selected DL models reported the best performance when trained with an SGD optimizer. A disappointing performance was noted when ResNet50 was trained with RMSprop and Adam. Comparable performance results can be seen from the rest of the data reported in Table 5.

### 3.5. Empirical Comparison with Existing Tools

Experimental comparisons were conducted with existing tools to establish the relevance of the proposed approach. We selected open-source optimization libraries implemented through known algorithms. Existing tools for auto-optimization were Hyperopt [21], Skopt [22], SMAC [23], and KerasTuner. Hyperopt uses Bayesian optimization and works best with classical ML models for various tasks other than computer vision. Similarly, Skopt and SMAC provide implementations suitable for ML tasks other than computer vision. For this reason, we chose KerasTuner from the selected optimization libraries.

KerasTuner optimizes DL models in two stages: first, it optimizes the hyperparameter search for a hyper model and returns optimal values. Second, it builds a model with optimal parameters and fits it into the training set. In our proposed approach, the first stage was executed when studying the association between hyperparameter values. Multiple experimental runs were conducted to reduce the bias of the reported results due to the stochastic nature of machine learning experiments. The average weight of accuracy and loss was registered to compare the performance of the existing tool and the proposed approach.

Figure 3 compares model performance when optimized using KerasTuner and the proposed approach. We first build and train a DL model with KerasTuner while observing the loss and accuracy for each experimental run. Then, we create a model with a similar architecture and optimize it with our proposed approach. The top half of the figure shows the model performance when optimized with KerasTuner. The bottom half of the figure offers model performance when optimized with our proposed approach.

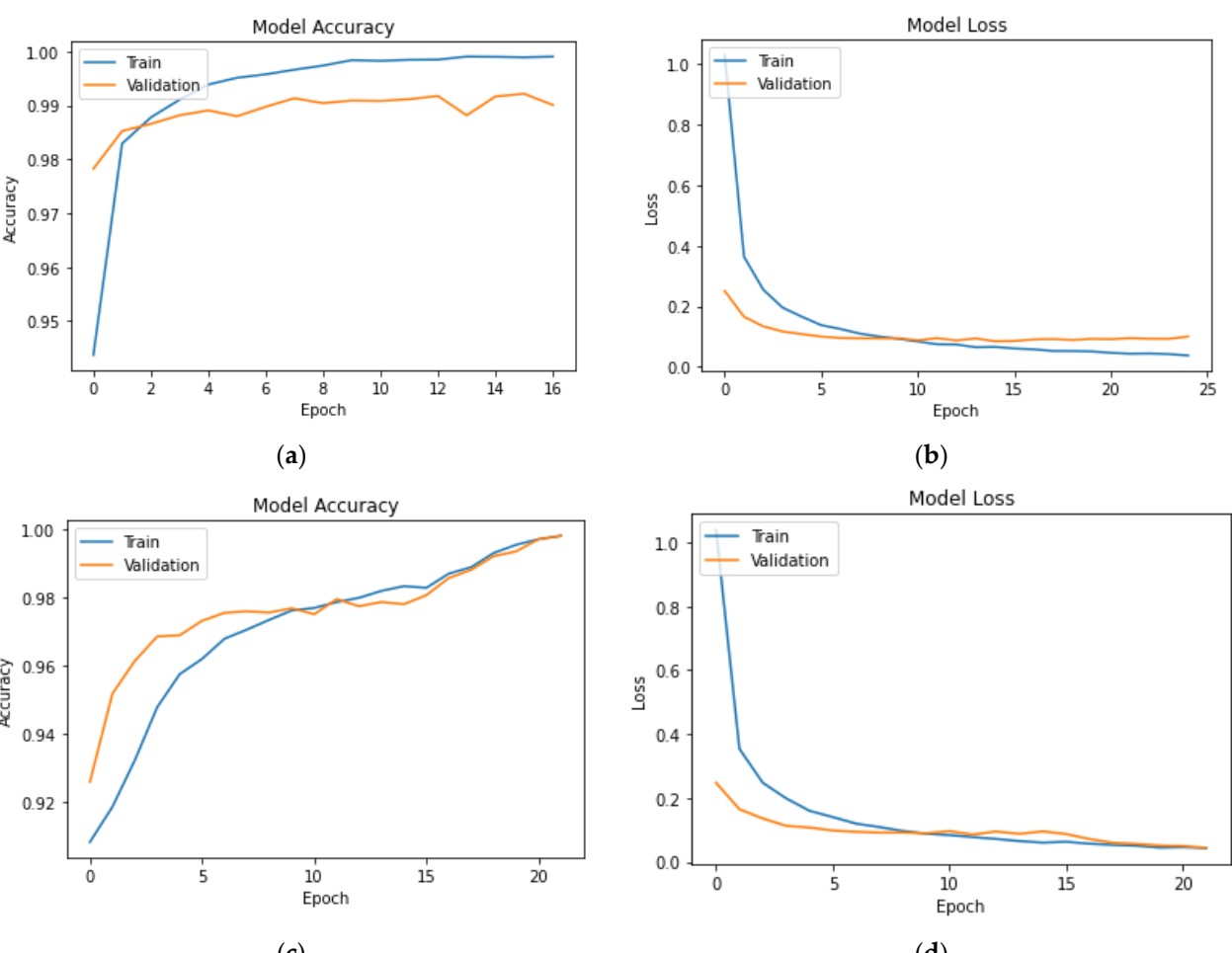

**Figure 3.** Model performance when optimized with KerasTuner and our proposed approach: (**a**) model accuracy when trained with KerasTuner; (**b**) model loss when trained with KerasTuner; (**c**) model accuracy when trained with our approach; (**d**) model loss when trained with our approach.

As can be seen from Figure 3a, the model quickly converges to around 98.5% accuracy and diverges when optimized using KerasTuner. Optimizing with our proposed approach reveals a convergence at approximately 99.7%, as seen in Figure 3b,c, which presents the model loss as it converges at around 10% before overfitting when optimized with KerasTuner. The model loss in Figure 3d converges at about 5% when optimized with our proposed approach.

From the result in Figure 3, it is apparent that the proposed approach improves the performance of DL when compared with the existing library. There is an increase of 1.2% in model accuracy and a decrease of 5% in loss after utilizing our proposed approach. The proposed solution is essential in building and optimizing DL models for image classification tasks.

### 3.6. Optimization of Transformer-Based Models

The proposed optimization solution was applied to a transformer-based architecture to verify its applicability. The ViT model was trained for 100 epochs while monitoring accuracy and loss. Figure 4 presents the model performance.

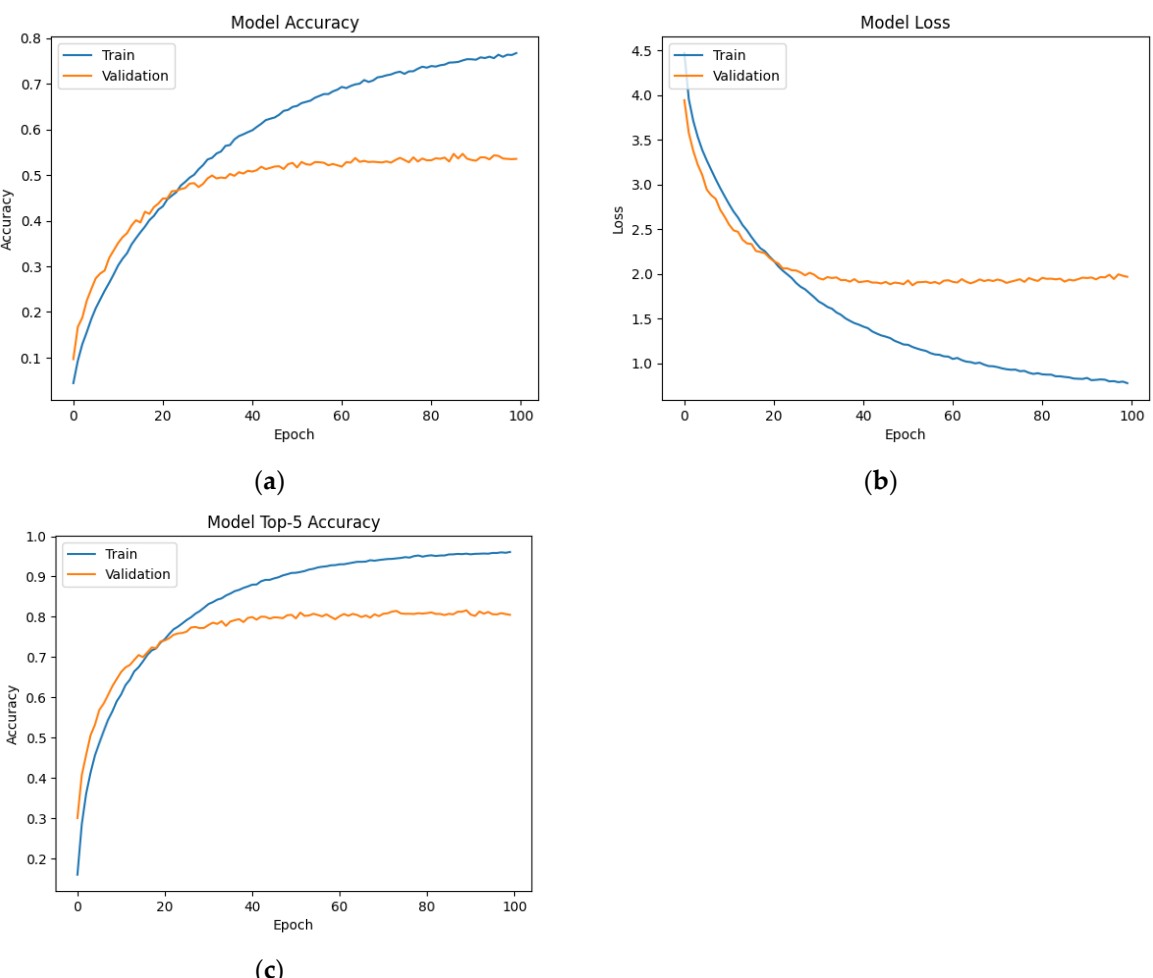

**Figure 4.** Transformer-based model performance when optimized with our proposed approach: (**a**) model accuracy; (**b**) model loss; (**c**) model top-5 accuracy.

The data in Figure 4a show that model accuracy converges at around 50% after 20 training iterations before overfitting. Training loss and validation loss improved to about 20%; as the training loss improved, the validation loss plateaued, as seen in Figure 4b.

The result in Figure 4 suggests that transformer-based models should be trained on larger image datasets for many iterations to ensure convergence. Current experiments used

small datasets like Fashion-MNIST and Stanford Cars. The training iteration was fixed to 100 to provide a fair comparison with ConvNet-based models. This accounts for the suboptimal results observed in the ViT model.

## 4. Discussion

This section discusses the results presented in the previous section and positions our contributions relative to knowledge in the existing literature.

### 4.1. Contributions to Related Literature

Probst et al. [40] employed a statistical approach to study the importance of hyperparameter tuning and introduce tunability, signifying the gain in the model's performance achieved through hyperparameter tuning. However, the research paper was confined to six classical machine learning models.

Faes et al. [41] researched the practicability of automated deep learning design by medical practitioners with non-programming and DL expertise. Developed models through the automated stream were compared with SOTA DL models on medical image classification tasks using publicly available datasets. Results indicated comparable discriminative performance and diagnostic properties [41].

Xu et al. [42] introduced an automatic network adaptation framework for object detection that goes beyond searching a classification backbone. The authors proposed the Auto-FPN framework with two modules performing an auto search: Auto-Fusion and Auto-Head. For both modules, the search space of gradient-based architecture claimed to be efficient with resource constraints [42].

Weerts et al. [43] introduced turning risk, an incurred loss into performance when a hyperparameter is not tuned but left to default value. Authors asserted that a model with default values might even outperform a model with adjustments to its hyperparameter in some cases [43]. However, only two classical ML algorithms were experimented with: Support Vector Machine and Random Forest.

Guo et al. [44] proposed a hierarchical trinity search framework that automatically discovers effective architecture for object detection. Their solution employs an end-to-end approach to discover all components and architectures of object detection simultaneously. With much less computational overhead, experimental results suggest that the proposed framework outperforms both manually designed and NAS-based architectures [44].

X. He, Wang, et al. [45] conducted a series of empirical evaluations to benchmark DL models and automated model design for COVID-19 detection. A random search was proven to deliver DL models with SOTA performance results through empirical results [45]. However, automating the optimization process of selected architectures after NAS remains not to be seen.

Yi and Bui [46] presented a deep learning model with an automated hypermeter search for highway traffic prediction. Their work employed a long short-term memory (LSTM) model and a recurrent neural network (RNN) variant to build a time series model for an intelligent transport system. Bayesian optimization is claimed to be superior to a manual search, grid search, or random search in searching for hyperparameter configuration [46].

While the literature on AutoML already exists, to our knowledge, it is limited to automating the design on DL models. Table 6 summarizes the research contribution to the existing body of knowledge. Thus far, some authors have attempted to highlight the significance of hyperparameter optimization [40,43]. Other researchers are more concerned with automating the design of deep learning models [41,42,44,45], while [46] focuses on automating the hyperparameter search for time series deep learning models.

Based on the reported literature, this paper concentrates on automating the optimization of deep learning models for image classification tasks. Experimental work examines the importance of hyperparameters and searches for the best value by automating the optimization process. Similarly, it empirically demonstrates that the proposed approach can improve the performance of deep learning models through automation.

**Table 6.** Research contributions of existing literature.

| Reference | Approach | ML Model(s) | Contribution |
|:---:|:---:|:---:|:---:|
| [40,43] | Manual | Classical | Investigated the importance of hyperparameter tuning |
| [41,42,44,45] | Auto | Deep Learning | Demonstrated the automated design of DL models |
| [46] | Auto | Deep Learning (Time Series) | Demonstrated the automated hyperparameter search |
| Our Approach | Auto | Deep Learning (Image Classification) | Investigated hyperparameter association, demonstrated automated optimization, proposed hyperparameter values |

*4.2. Association between Batch Size and Learning Rate*

Our results suggest a significant positive correlation between batch sizes and learning rates. This implies that lower rates should be used with smaller batches for better results, and the inverse is true for larger batches. Following the present results, previous studies have suggested using smaller batches [47,48] and high learning rates with large batch sizes [49]. Furthermore, batch size 32 was recommended as the default value that provides good results [50].

Another critical finding distinctive from the previously reported results is observed when a learning rate of 0.1 is selected. Regardless of how the batch size was iterated in all DL models, validation accuracy was at around 50%. An implication of this finding is the possibility that high rates plateau the performance close to a specific value. However, with the stochastic nature of ML experiments, these results must be interpreted cautiously.

*4.3. Optimization Implications*

The post-automated optimization results signify that the proposed scheme can improve the performance of DL models on a given image classification task. VGG-16 consumed approximately an hour of training and adjusting the knobs to reduce the error rate by 73%. Likewise, for EfficientNet, optimizing to a final accuracy took 2 h and 41 min.

Without automated optimization tools, data scientists spend hours manually training and optimizing DL models. Additionally, the optimization process requires a good knowledge of the impact of both model- and training-related parameters relative to performance evaluation metrics. With such constraints, it is evident that building and optimizing DL for image classification becomes a burdensome task.

However, with a fully automated machine learning pipeline, data scientists shall be liberated from the majority of burdensome tasks involved in deploying DL models. For this reason, the proposed automated optimization scheme becomes significant. Data scientists can leverage such a tool during the design and optimization process, consequently becoming more productive and efficient in completing other valuable scientific tasks in the machine learning pipeline. This implies that data scientists will only be required to define a task, model, and dataset and then submit it to automated machine learning tools.

*4.4. Proposed Approach on Vision Transformers*

The proposed tool can be applied to fine-tune and optimize transformer-based models automatically. However, it has to be customized to incorporate ViT hyperparameters. ViT hyperparameters include patch size, projection dimension, number of heads, number of transformer layers, multilayer perceptron units, and the standard convent parameters. This implies that the search space will be more extensive when compared to ConvNets.

Transformer-based models must be trained on larger datasets for long enough to ensure convergence [51]. The number of training iterations can be set to 500 with early stopping to control overfitting. This signifies that ViT models can be trained for up to five times more iterations than ConvNet models.

Similarly, transformer-based models can be trained with large batch sizes compared with convent models. Our results suggest that mini batches between 16 and 64 promise optimal performance with convents, but ViT models can be trained with 256 and 512 batches. The results presented in Section 3.6 were obtained when the ViT model was trained using a batch size of 256.

The proposed approach can be customized to train and optimize ViT models. The search space must be expanded to accommodate transformer-based hyperparameters. Similarly, training time must be long enough, with many iterations to guarantee convergence to an optimal solution.

## 5. Conclusions

The main goal of this study was to amplify the understanding of AutoML, and especially the optimization of DL models for image classification tasks. We have proposed a framework to automate the optimization process of DL models. Four DL models and prominent datasets for image classification challenges were selected for our empirical evaluations. We extended our experiments to include a transformer-based model for image classification. Our practical tests explored the association of two significant training-related parameters: batch size and learning rate. We have demonstrated that the proposed scheme improves the performance of DL models and significantly reduces the error rate by up to a 73% relative drop. Similarly, we have presented the training-related parameter values most likely to result in the improved performance of DL models for further empirical evaluations. The particular contribution of this paper is that it introduces a tool to automate the optimization process of DL models for image classification tasks. Instead of using brute force and exhaustive searches, a data scientist controls the number optimization attempt in the proposed framework. Image classification is considered as an ancient AI problem; nevertheless, it remains worthy of attention, as an image still speaks more than a thousand words. More research work will need to be done in efforts to fully automate these AI task's ML pipeline. The insights gained from this study may assist in efforts to radically democratize the building and optimization of DL models, even for amateurs. While computer vision tasks involve image classification, object detection, and segmentation, implementation and empirical evaluations of the proposed solution are limited to image classification. Similarly, a data scientist must specify a preferred DL model for a given classification task before the optimization process. This demands a basic knowledge of existing or custom ConvNet-based models. In the future, we will seek to improve and deliver our automated optimization solution, which can be used as a Python library. Similarly, developing a web interface without additional package installation requirements allows non-practitioners to experiment and test the solution.

**Author Contributions:** Conceptualization, D.M.M.; methodology, D.M.M. and D.G.N.; formal analysis, D.M.M. and D.G.N.; writing—original draft preparation, D.M.M.; writing—review and editing, D.G.N., S.S. and S.K.; supervision, D.G.N., S.S. and S.K. All authors have read and agreed to the published version of the manuscript.

**Funding:** This work was carried out with the aid of a grant from the Artificial Intelligence for Development in Africa Program, a program funded by the Canada's International Development Research Centre, Ottawa, Canada and the Swedish International Development Cooperation Agency, grant number 109704-001/002. The APC was funded by AI4D Anglophone Africa Multidisciplinary Research Lab.

**Institutional Review Board Statement:** Not applicable.

**Informed Consent Statement:** Not applicable.

**Data Availability Statement:** Codes for the empirical test can be accessed from Google drive: https://drive.google.com/drive/folders/1G8uBoBp6hpSr9MeleeuFCVIEqKC5EOwF?usp=share_link, accessed on 7 August 2023; Fashion-MNIST dataset: https://github.com/zalandoresearch/fashion-mnist, accessed on 29 June 2023; Stanford Cars dataset: https://ai.stanford.edu/~jkrause/cars/car_

**Conflicts of Interest:** The authors declare no conflict of interest.

## Abbreviations

The following abbreviations are used in this manuscript:

| | |
|---|---|
| AutoML | Automated Machine Learning |
| CIFAR | Canadian Institute for Advanced Research |
| ConvNet | Convolutional Neural Networks |
| CV | Computer Vision |
| DL | Deep Learning |
| ML | Machine Learning |
| MNIST | Modified National Institute of Standards and Technology |
| RMSProp | Root Mean Squared Propagation |
| SGD | Stochastic Gradient Descent |
| SMBO | Sequential-Model-Based Optimization |
| SOTA | State-of-the-Art |
| ViT | Vision Transformer |
| VGG | Virtual Geometry Group |

## Appendix A

**Table A1.** Training data for VGG-16.

| Batch Size | Learning Rate | Loss | Accuracy | Validation Loss | Validation Accuracy | Epoch | Time (min) |
|---|---|---|---|---|---|---|---|
| 8 | 0.1 | 0.8173 | 0.4593 | 0.7953 | 0.4592 | 15 | 10 |
| 16 | 0.1 | 1.6638 | 0.4641 | 0.7969 | 0.4611 | 12 | 7 |
| 32 | 0.1 | 0.7983 | 0.4692 | 0.7938 | 0.4607 | 11 | 5 |
| 64 | 0.1 | 0.7955 | 0.4517 | 0.791 | 0.4612 | 10 | 7 |
| 128 | 0.1 | 0.7922 | 0.4674 | 0.7942 | 0.4615 | 15 | 8 |
| 8 | 0.01 | 0.6868 | 0.7148 | 0.3437 | 0.8288 | 32 | 20 |
| 16 | 0.01 | 0.5479 | 0.7364 | 0.4557 | 0.8245 | 27 | 17 |
| 32 | 0.01 | 0.488 | 0.7377 | 0.2931 | 0.8569 | 26 | 15 |
| 64 | 0.01 | 0.4046 | 0.7587 | 0.3713 | 0.8583 | 13 | 9 |
| 128 | 0.01 | 0.3861 | 0.7728 | 0.2908 | 0.8734 | 16 | 8 |
| 8 | 0.001 | 0.3983 | 0.8046 | 0.2417 | 0.8754 | 19 | 13 |
| 16 | 0.001 | 0.32 | 0.7812 | 0.2482 | 0.8672 | 21 | 12 |
| 32 | 0.001 | 0.3387 | 0.8232 | 0.2407 | 0.862 | 23 | 15 |
| 64 | 0.001 | 0.2725 | 0.8463 | 0.2318 | 0.8848 | 29 | 18 |
| 128 | 0.001 | 0.2716 | 0.8546 | 0.2368 | 0.8867 | 22 | 14 |
| 8 | 0.0001 | 0.2976 | 0.8514 | 0.2292 | 0.8925 | 29 | 17 |
| 16 | 0.0001 | 0.2524 | 0.8714 | 0.3784 | 0.8526 | 23 | 13 |
| 32 | 0.0001 | 0.2171 | 0.8784 | 0.1949 | 0.9038 | 13 | 8 |
| 64 | 0.0001 | 0.2229 | 0.8811 | 0.1945 | 0.9071 | 21 | 10 |
| 128 | 0.0001 | 0.2113 | 0.8879 | 0.1971 | 0.9107 | 6 | 11 |

**Table A2.** Training data for Inception.

| Batch Size | Learning Rate | Loss | Accuracy | Validation Loss | Validation Accuracy | Epoch | Time (min) |
|---|---|---|---|---|---|---|---|
| 8 | 0.1 | 10.4708 | 0.4874 | 0.8036 | 0.4682 | 29 | 10 |
| 16 | 0.1 | 0.8075 | 0.4743 | 0.7765 | 0.4825 | 11 | 3 |
| 32 | 0.1 | 0.8416 | 0.4871 | 0.772 | 0.4927 | 11 | 3 |
| 64 | 0.1 | 0.795 | 0.4823 | 0.7636 | 0.4837 | 19 | 5 |
| 128 | 0.1 | 0.841 | 0.4741 | 0.7928 | 0.4715 | 23 | 7 |
| 8 | 0.01 | 0.5846 | 0.7213 | 0.2506 | 0.7905 | 18 | 6 |
| 16 | 0.01 | 0.6324 | 0.6924 | 0.708 | 0.8738 | 12 | 3 |

**Table A2.** *Cont.*

| Batch Size | Learning Rate | Loss | Accuracy | Validation Loss | Validation Accuracy | Epoch | Time (min) |
|---|---|---|---|---|---|---|---|
| 32 | 0.01 | 0.3362 | 0.8246 | 0.1505 | 0.9005 | 16 | 5 |
| 64 | 0.01 | 0.2596 | 0.8623 | 0.14 | 0.9056 | 22 | 6 |
| 128 | 0.01 | 0.1719 | 0.8959 | 0.1472 | 0.909 | 11 | 3 |
| 8 | 0.001 | 0.1709 | 0.9022 | 0.1589 | 0.9067 | 33 | 11 |
| 16 | 0.001 | 0.1336 | 0.9097 | 0.1093 | 0.924 | 24 | 8 |
| 32 | 0.001 | 0.1036 | 0.9301 | 0.0986 | 0.926 | 17 | 5 |
| 64 | 0.001 | 0.0978 | 0.9283 | 0.1078 | 0.9183 | 11 | 3 |
| 128 | 0.001 | 0.1213 | 0.919 | 0.1187 | 0.9261 | 23 | 6 |
| 8 | 0.0001 | 0.1304 | 0.9203 | 0.1278 | 0.9267 | 37 | 12 |
| 16 | 0.0001 | 0.1045 | 0.9288 | 0.0908 | 0.9376 | 25 | 7 |
| 32 | 0.0001 | 0.067 | 0.9419 | 0.0879 | 0.9378 | 46 | 11 |
| 64 | 0.0001 | 0.1298 | 0.9247 | 0.0991 | 0.9375 | 21 | 7 |
| 128 | 0.0001 | 0.1004 | 0.9363 | 0.1062 | 0.9334 | 11 | 4 |

**Table A3.** Training data for EfficientNetB0.

| Batch Size | Learning Rate | Loss | Accuracy | Validation Loss | Validation Accuracy | Epoch | Time (min) |
|---|---|---|---|---|---|---|---|
| 8 | 0.1 | Nan | 0.5000 | Nan | 0.5000 | 11 | 7 |
| 16 | 0.1 | Nan | 0.5000 | Nan | 0.5000 | 11 | 6 |
| 32 | 0.1 | Nan | 0.5000 | Nan | 0.5000 | 10 | 6 |
| 64 | 0.1 | Nan | 0.5000 | Nan | 0.5000 | 10 | 6 |
| 128 | 0.1 | Nan | 0.5000 | Nan | 0.5000 | 10 | 6 |
| 8 | 0.01 | Nan | 0.7407 | Nan | 0.8742 | 11 | 6 |
| 16 | 0.01 | 1264.5101 | 0.8159 | 158.342 | 0.8877 | 11 | 6 |
| 32 | 0.01 | 0.1818 | 0.8916 | 0.0885 | 0.9213 | 29 | 20 |
| 64 | 0.01 | 0.151 | 0.859 | 3.329 | 0.921 | 18 | 9 |
| 128 | 0.01 | 8696.4881 | 0.8901 | 145.0845 | 0.9306 | 32 | 16 |
| 8 | 0.001 | 0.0934 | 0.9253 | 0.0552 | 0.9329 | 17 | 10 |
| 16 | 0.001 | 0.0638 | 0.9374 | 0.0491 | 0.9369 | 21 | 10 |
| 32 | 0.001 | 0.0605 | 0.9374 | 0.0492 | 0.9421 | 17 | 8 |
| 64 | 0.001 | 0.0744 | 0.9342 | 0.0511 | 0.9441 | 24 | 13 |
| 128 | 0.001 | 0.0764 | 0.9345 | 0.0483 | 0.9467 | 12 | 7 |
| 8 | 0.0001 | 0.1003 | 0.9279 | 0.0488 | 0.9485 | 16 | 10 |
| 16 | 0.0001 | 0.065 | 0.9447 | 0.0487 | 0.9486 | 30 | 14 |
| 32 | 0.0001 | 0.0702 | 0.9427 | 0.0461 | 0.9487 | 11 | 5 |
| 64 | 0.0001 | 0.0734 | 0.9436 | 0.0985 | 0.9496 | 25 | 13 |
| 128 | 0.0001 | 0.0773 | 0.9505 | 0.0425 | 0.9546 | 11 | 5 |

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
