# Peer review of "Automated Optimization-Based Deep Learning Models for Image Classification Tasks"

_computers, doi:10.3390/computers12090174_

Round 1

Reviewer 1 Report

The authors propose an automated optimization scheme for deep learning models in image classification tasks. Using a sequential model-based optimization algorithm, they demonstrate improved performance, achieving a 73% relative error rate drop on the cats and dogs' dataset within an hour of automated optimization, and also propose training-related parameter values for better model performance.

  1. The authors should use other datasets like the Fashion-MNIST and the Stanford Cars dataset to demonstrate that these techniques generalize well. The benefit of AutoML based techniques can only be demonstrated on real world datasets and not just benchmark datasets.  Fashion-MNIST and the Stanford Cars provide slightly more real-world versions of the image classification datasets.
  1. The authors do not use transformer based models although they have provided some of the state of the art models alongside some convent models like EfficientNet (which the authors use). Ideally, the authors should provide these experiments for at least one image transformer based model. But, at least the authors should change the title to clarify that their work is limited to convnet models. 

  2. The authors should consider creating a website so that non-practitioners can test out this on small datasets. 

There are quite a few language issues. It should be run through a grammar check before publication. 

Author Response

Response to Reviewer 1 Comments

Point 1: The authors should use other datasets like the Fashion-MNIST and the Stanford Cars dataset to demonstrate that these techniques generalize well. The benefit of AutoML based techniques can only be demonstrated on real world datasets and not just benchmark datasets.  Fashion-MNIST and the Stanford Cars provide slightly more real-world versions of the image classification datasets.

Response 1: Fashion-MNIST and Stanford Cars datasets were featured in the experiments per the reviewer’s suggestion. Consequently, MNIST and CIFAR datasets were removed from the experiments. A positive correlation between batch size and learning was observed with the introduced datasets. Also, observation of evaluation metrics (accuracy, validation accuracy) revealed a slight decrease of around 7 – 10.9% when changing the dataset, for instance from MNIST to Fashion-MNIST.

For that reason, it can be concluded that the proposed approach works well on real word datasets, albeit a slight change in the selected evaluation metric may be observed.

Point 2: The authors do not use transformer-based models although they have provided some of the state-of-the-art models alongside some convent models like EfficientNet (which the authors use). Ideally, the authors should provide these experiments for at least one image transformer-based model. But at least the authors should change the title to clarify that their work is limited to convnet models.

Response 2: A Vision Transformer model was included in experimenting with the proposed solution.

A transformer-based model by Dosovitskiy et al. was used to test the applicability of the proposed approach to Vision Transformers (ViT). The model contains multiple transformer blocks with a multi-head attention layer applied to the sequence of image patches. The final output of each transformer block is flattened to be used as the representation of image input to the classifier head. The classifier head uses a softmax activation to produce the output probability of each class.

The result shows that model accuracy converges at around 50% after 20 training iterations before overfitting. Training loss and validation loss improved to around 20%, then as the training loss improves the validation loss plateaued. The result in Figure 4 suggests that transformer-based models should be trained on larger image datasets for many iterations to ensure convergence. Current experiments used small datasets like fashion-MNIST and Stanford cars. Also, the training iteration was fixed to 100 to provide a fair comparison with Convnet-based models. This accounts for the suboptimal results observed in the ViT model.

Taken together, the proposed approach can be customized to train and optimize ViT models. The search space must be expanded to accommodate transformer-based hyperparameters. Also, training time must be long enough with many iterations to guarantee convergence to an optimal solution.

Point 3: The authors should consider creating a website so that non-practitioners can test out this on small datasets.

Response 3: The recommendation has been taken and featured in our future works. A web interface embedded with the proposed tool shall provide an avenue to a diverse audience including non-practitioners to experiment and test the optimization tool.

Reviewer 2 Report

This manuscript proposes an automated optimization scheme for deep learning models in image classification tasks, aiming to alleviate the reliance on human expertise and the labor-intensive nature of the optimization process. The manuscript presents some improvement results, and also suggests training-related parameter values to enhance the performance of deep learning models, offering the potential to free data scientists from manual and burdensome optimization tasks. However, I have below primary comments concerned:

  1. The author should provide a more focused summary of their contributions, methodology, and results in abstract.
  2. The manuscript in abstract states that the optimization process relies on human expertise and is labor-intensive, but fails to provide specific details or references to support this claim.
  3. In section 2.7, it mentions the proposal of an automated optimization scheme for deep learning models, but does not clearly explain the novelty or uniqueness of this approach compared to existing literature. Figure 1 is unclear.
  4. In section 3.1, it claims that the proposed scheme improves the, but does not provide sufficient evidence to support this claim and auto optimization features. The improvement they showed is very limited in Table 3.
  5. The manuscript mentions the proposal of training-related parameter values for better performance, but does not provide specific details on these values. Figure 2 is not sufficient to show the auto optimization feature.
  6. The manuscript concludes by stating that the proposed scheme may free data scientists from manual and onerous optimization tasks, but does not elaborate on how this would be achieved.

Overall, I suggest reject this manuscript.

The abstract would benefit from proofreading and editing for grammar and clarity. There are several instances where the wording is awkward or unclear, making it difficult to understand the intended meaning.

Author Response

Response to Reviewer 2 Comments

Point 1: The author should provide a more focused summary of their contributions, methodology, and results in the abstract.

Response 1: The abstract was reviewed.

Applying deep learning models requires design and optimization when solving multifaceted Artificial Intelligence tasks. Optimization relies on human expertise and is achieved only with great exertion. Current literature concentrates on automating the design, leaving insufficient work on the optimization. Also, most existing optimization libraries concentrate on other machine-learning tasks rather than image classification. For that reason, an automated optimization scheme of deep learning models for image classification tasks is proposed in this paper. A sequential model-based optimization algorithm was used to implement the proposed scheme. Four deep-learning models, a transformer-based model, and common datasets for image classification challenges were employed in the experiments. Through empirical evaluations, this paper demonstrates that the proposed scheme improves the performance of deep learning models. In particular, for a VGG-16, accuracy was heightened from 0.937 to 0.983 signifying a 73% relative error rate drop within an hour of automated optimization. Similarly, training-related parameter values are proposed for better performance of deep learning models. The scheme can be extended to automate the optimization of transformer-based models. The insights gained from this study may assist efforts to radically democratize the building and optimization of DL models, even for amateurs.

Point 2: The manuscript in the abstract states that the optimization process relies on human expertise and is labor-intensive, but fails to provide specific details or references to support this claim.

Response 2: existing literature documents that optimizing machine learning models principally relies on human expertise and is often a labor-intensive process [1].

[1]        M. Claesen and B. De Moor, “Hyperparameter Search in Machine Learning,” Feb. 2015, [Online]. Available: http://arxiv.org/abs/1502.02127

Point 3: In section 2.7, it mentions the proposal of an automated optimization scheme for deep learning models, but does not clearly explain the novelty or uniqueness of this approach compared to existing literature. Figure 1 is unclear.

Response 3: The proposed solution summarized by a scheme in section 2.7 can be differentiated from existing literature in three ways. First, it automates the optimization process of models for computer vision, a solution currently insufficient in the existing literature. The majority of existing literature concentrates on optimizing models for other tasks rather than computer vision.

Second, it supports state-of-the-art deep learning models with transfer learning for any image classification task. Existing literature is dominated with solution that features classical ML models and design of deep learning models, leaving insufficient works on optimization process. Third, a user controls the optimization process and may limit utilization of resource, contrary to brute force and exhaustive search.

Figure 1 provides a framework that summarizes the entire proposed automated optimization process. A user specifies the DL model, dataset for classification task, and an optimization attempt control variable as inputs. After the automated process, final best model, configurations, and evaluation metrics is provided as outputs.

Point 4: In section 3.1, it claims that the proposed scheme improves the, but does not provide sufficient evidence to support this claim and auto optimization features. The improvement they showed is very limited in Table 3.

Response 4: Section 3.2 presents results of the performance evaluation after training and optimization by using the proposed automated scheme. Table 3 provides the evaluation of optimization process by using the proposed scheme in terms of accuracy and training time. Accuracy represents model performance after the automated optimization process as an evaluation criterion. Training time represents the consumption of resources during the automated optimization process.

Improvement in model performance as a result of the proposed automated scheme can be witnessed on the reduction of error rate as accuracy improves. In a similar fashion, improvements on other evaluation metrics like the loss after the automated optimization process shows the significance of the proposed scheme on model performance. Table 4 presents the results evaluation of the training loss.

Point 5: The manuscript mentions the proposal of training-related parameter values for better performance, but does not provide specific details on these values. Figure 2 is not sufficient to show the auto optimization feature.

Response 5: The proposed values can be adopted as a pair of default parameters during the process of training DL for image classification tasks. Based on our empirical evaluations, these values guarantee an optimal performance of DL models. Data scientist and DL practitioners may select and apply a pair of proposed values when building a baseline model. After-wards, the optimization process in efforts to surpass a baseline model may be initiated by utilizing automated tools like the proposed scheme.

Figure 2 provides experimental results to support the established relationship between batch size and learning rate and their impact on both model accuracy and loss. The figure entails that accuracy and loss improves positively when the model is trained using low rates and small minibatch

Point 6: The manuscript concludes by stating that the proposed scheme may free data scientists from manual and onerous optimization tasks, but does not elaborate on how this would be achieved.

Response 6: Without such tools for automated machine learning, building and optimizing models will principally rely on human expertise. During such process, data scientists will have to spend reasonable amount of time to manually adjust parameters in seeking for a better performance. Also, the process requires a good knowledge on the association and impact of both model-related and training-related parameters on the evaluation metric.

However, tools to automate the machine learning pipeline, like the proposed scheme, may be applied during the optimization process. For that reason, a task that would have required a manual interaction with human can be achieved by applying automated tools. This implies, data scientists will only be required to define a task, model, and dataset and then submit it to automated machine learning tools. [Section 4.2 provides an explanation on the comment]

Reviewer 3 Report

The paper explores the optimization of network architectures using AutoML and introduces a novel solution developed by the authors.

The introduction of the paper effectively provides context for the problem and discusses other AutoML tools, including NAS, which is relevant to the topic.

Regrettably, the authors do not offer a quantitative comparison of their approach to these existing methods in this paper. It remains unclear what advantages their approach offers compared to other AutoML frameworks that provide similar tools for similar studies.

Furthermore, the authors do not draw any architectural conclusions concerning AutoML. The number of investigated problems is quite limited, and the examined problems and architectures are outdated. It would be beneficial for them to consider more recent problems, such as object detection and segmentation in COCO and visual transformers. Most architectures perform exceptionally well on MNIST and CIFAR, making these experiments less relevant within such a framework, as hyperparameters have no significant effect.

Based on these shortcomings, I cannot wholeheartedly recommend accepting the paper. The only comparison with other methods is found in section 4.1, which is solely qualitative and lacks a quantitative analysis. The authors should address these issues to improve the chances of acceptance.

Author Response

Response to Reviewer 3 Comments

Point 1: Regrettably, the authors do not offer a quantitative comparison of their approach to these existing methods in this paper. It remains unclear what advantages their approach offers compared to other AutoML frameworks that provide similar tools for similar studies.

Response 1: Section 3.5 has been added to include empirical comparison with existing tools from similar studies. We selected open-source optimization libraries implemented through known algorithms. Existing tools for auto-optimization were Hyperopt, Skopt, SMAC, and KerasTuner.

Hyperopt uses Bayesian optimization and works most with classical ML models for various tasks rather than computer vision. In a similar fashion, Skopt and SMAC pro-vides implementations suitable for ML tasks rather than computer vision. For that reason, we experimented with KerasTuner from the selected optimization libraries.

KerasTuner optimizes DL models in two stages: first, it optimizes the hyperparameter search for hyper model and returns optimal values. Second, it builds the model with optimal parameters and fits it on the training set. In our proposed approach, the first stage is executed when studying the association between hyperparameter values.  We first build and train a DL with KerasTuner for a fair comparison while observing the loss and accuracy. Then, we build a model with similar architecture and optimize it with our proposed approach.

the model quickly converges to around 87% accuracy and diverges when optimized using KerasTuner. Optimizing with our proposed approach reveals a convergence at around 97% accuracy. Also, training loss converged at around 35% with KerasTuner and at around 10% with our proposed approach.

From the result in Figure 3, it is apparent that the proposed approach improves the performance of DL compared with the existing library. There is an increase of 10% in model accuracy and a decrease of 25% in loss after utilizing our proposed approach. The proposed solution is essential in building and optimizing DL models for image classification tasks.

Point 2: Furthermore, the authors do not draw any architectural conclusions concerning AutoML. The number of investigated problems is quite limited, and the examined problems and architectures are outdated. It would be beneficial for them to consider more recent problems, such as object detection and segmentation in COCO and visual transformers. Most architectures perform exceptionally well on MNIST and CIFAR, making these experiments less relevant within such a framework, as hyperparameters have no significant effect.

Response 2:

Fashion-MNIST and Stanford Cars datasets were featured in the experiments to provide real-world versions of image classification tasks. Consequently, MNIST and CIFAR datasets were removed from the experiments. A positive correlation between batch size and learning was observed with the introduced datasets. Also, observation of evaluation metrics (accuracy, validation accuracy) revealed a slight decrease of around 7 – 10.9% when changing the dataset, for instance from MNIST to Fashion-MNIST. For that reason, it can be concluded that the proposed approach works well on real word datasets, albeit a slight change in the selected evaluation metric may be observed.

A Vision Transformer model was included in experimenting the proposed solution.

A transformer-based model by Dosovitskiy et al. was used to test the applicability of the proposed approach to Vision Transformers (ViT). The model contains multiple transformer blocks with a multi-head attention layer applied to the sequence of image patches. The final output of each transformer block is flattened to be used as the representation of image input to the classifier head. The classifier head uses a softmax activation to produce the output probability of each class.

The result shows that model accuracy converges at around 50% after 20 training iterations before overfitting. Training loss and validation loss improved to around 20%, then as the training loss improves the validation loss plateaued. The result in Figure 4 suggests that transformer-based models should be trained on larger image datasets for many iterations to ensure convergence. Current experiments used small datasets like fashion-MNIST and Stanford cars. Also, the training iteration was fixed to 100 to provide a fair comparison with Convnet-based models. This accounts for the suboptimal results observed in the ViT model.

Taken together, the proposed approach can be customized to train and optimize ViT models. The search space must be expanded to accommodate transformer-based hyperparameters. Also, training time must be long enough with many iterations to guarantee convergence to an optimal solution.

The proposed solution concentrated on image classification tasks since it is an essential component of our ongoing research. However, the tool can be improved and extended to detection and segmentation as part of future works. Therefore, we aim to demonstrate the applicability of proposed automated optimization tool in image classification task as a benchmark.

Point 3: Based on these shortcomings, I cannot wholeheartedly recommend accepting the paper. The only comparison with other methods is found in section 4.1, which is solely qualitative and lacks a quantitative analysis. The authors should address these issues to improve the chances of acceptance.

Response 3: comments and recommendations have been taken and resolved. A separate section to present empirical comparison (Section 3.5) is added. The proposed approach has been extended to Vision Transformer architectures to test its applicability. Also, additional datasets were included to provide real-life benchmarking of the proposed automated optimization approach.

Reviewer 4 Report

In this paper, the authors consider tools for automated optimization of deep learning models in case of image classification. They view their contributions as proposing a framework of automated optimization of deep-learning model and an empirical demonstration that the model performs well in real world conditions. They explore two training-related parameters - the batch size and the learning rate - and make recommendations for training-related parameter values for better performance.

This is the second version of the submitted paper and extensive corrections have been made, which, in my opinion improve the presentation. However, I did not have access to the reviews of the original paper and to the authors' response. 

Hence, my review is based on the paper as is.

The paper is well-structured. It starts with an introduction covering previous works. At the end of the introduction, the structure of the paper is given. The next section explains the methods, including the problem definition, the experimental setup, and the optimization algorithm. Then, the results are commented in a separate section. Finally, in the Conclusion, some directions for future research are outlined. There are 51 references, many of which are recent. Several of them are from the last few years, which shows that the research is of interest to the scientific community. There are three appendices with data from the conducted experiments.

The results are not surprising, but rather a step forward. They may be of interest to the scientific community dealing with image classification.

MAIN SUGGESTIONS FOR IMPROVEMENT:

1) The English needs further improvement. Below I give some examples, which are not exhaustive.

2) The three tables in the appendices show that the numbers were changed. It is not clear to me why.

TECHNICAL REMARKS:

1) Line 15: "leaving insufficient work on the optimization." - paraphrase

2) Line 22: VGG-16 - define before using

3) Line 26: "may assist efforts to radically democratize the building and optimization of DL models" - explain what "democratize" means in this case

4) Lines 79-81: "Model’s performance can be affected considerably by the choice of these parameters, nevertheless finding good values is notably tough [20]" - "nevertheless" should be replaced by "even if" or another more appropriate word.

5) Line 95: "To date, only an inadequate number of works of literatures on the automated optimization of DL models have been identified." - instead of "inadequate" use "insufficient" or paraphrase

5) Lines 105, 109, 193: Avoid starting sentences with "also"

6) Figs. 1, 2, 3, and 4: All texts must be of the same size that matches the main text.

7) Line 293: "Most of existing approach optimizes..." - Maybe "Most of the existing approaches optimize..."?

See above.

Round 2

Reviewer 1 Report

--

Author Response

Suggestions and comments have been resolved as per the review round 1.

Reviewer 2 Report

The authors have resolve all my concerns.

I am delighted to present a review report highlighting the remarkable improvement in the author's English language skills. 

Author Response

Suggestions and comments have been resolved as per the review round 1. The quality of English has been further improved during the review round two.

Reviewer 3 Report

The quality of the paper has improved significantly since the last revision. Many grammatical errors were corrected and it is great that the authors have added comparison with other existing approaches.

But unfortunately this comparison is not scientific. The comparison in section 3.5 contains a single figure (Figure 3). Computer vision is part of ML so i do not understand why Hyperopt or SKOPT can not be used in the case of this problem. comparisons with more than one existing approach should be done.

Additionally it seems that the comparison was done investigating a single run only, which is severely biased considering the random parameters in the methods.

If one examines the initial accuracies in figure 3 at epoch 0 one sen easily see, that the validation accuracy start from 0.86 in case of KerasTuner and 0.92 in case of the proposed method, which depicts a serious bias in the experiments. 

The experiments has to be repeated using multiple methods for comparison, multiple runs, where the average accuracy loss values and their variance is also repeated.

Without these experiments it is extremely difficult to judge and compare the performance of the approach and the acceptance of the manuscript can not be supported.

-
